# Simulation of Electronic Waste Reverse Chains for the Sao Paulo Circular Economy: An Artificial Intelligence-Based Approach for Economic and Environmental Optimizations

**DOI:** 10.3390/s23229046

**Published:** 2023-11-08

**Authors:** Geraldo Cardoso de Oliveira Neto, Sidnei Alves de Araujo, Robson Aparecido Gomes, Dario Henrique Alliprandini, Fabio Richard Flausino, Marlene Amorim

**Affiliations:** 1Business Administration and Industrial Engineering Post-Graduation Program, FEI University, Sao Paulo 01525-000, Brazil; geraldo.prod@gmail.com (G.C.d.O.N.); dario.allip@fei.edu.br (D.H.A.); fabiorflausino@gmail.com (F.R.F.); 2Informatics and Knowledge Management Post-Graduation Program, Universidade Nove de Julho—UNINOVE, Sao Paulo 01525-000, Brazil; saraujo@uni9.pro.br (S.A.d.A.); robson.gomes10@uni9.edu.br (R.A.G.); 3Department of Economics, Management, Industrial Engineering and Tourism (DEGEIT) and GOVCOPP, University of Aveiro, 3810-193 Aveiro, Portugal

**Keywords:** reverse logistic, reverse chains, WEEE, industry 4.0 technology, simulation, artificial intelligence, eco-efficiency, circular economy

## Abstract

The objective of this study was to apply simulation and genetic algorithms for the economic and environmental optimization of the reverse network (manufacturers, waste managers, and recyclers in Sao Paulo, Brazil) of waste from electrical and electronic equipment (WEEE) to promote the circular economy. For the economic evaluation, the reduction in fuel, drivers, insurance, depreciation, maintenance, and charges was considered. For the environmental evaluation, the impact of abiotic, biotic, water, land, air, and greenhouse gases was measured. It was concluded that the optimized structure of the WEEE reverse chains for Sao Paulo, Brazil provided a reduction in the number of collections, thus making the most of cubage. It also generated economic and environmental gains, contributing to the strategic actions of the circular economy. Therefore, the proposed approach is replicable in organizational practice, which is mainly required to meet the 2030 agenda of reducing the carbon footprint generated by transport in large cities. Thus, this study can guide companies in structuring the reverse WEEE chains in Sao Paulo, Brazil, and other states and countries for economic and environmental optimization, which is an aspect of great relevance considering the exponential generation of WEEE.

## 1. Introduction

Growing industrialization and increased competitiveness have contributed to the growth in the volume of electronic products that are manufactured across a variety of market segments. Currently, large volumes of electronic products are sold worldwide. As a result of this growth, electronic waste has become a major problem in the disposal process, presenting itself as a critical and aggressive situation for the environment [1]. This challenge triggered in Brazil in 2010 Law 12.305/2010, which was enacted on the National Solid Waste Policy. This law has been acknowledged as the most specific to concern reverse logistics and the recycling of WEEE. Its principle is to develop reverse chains to promote a shared responsibility for the life cycle of electrical and electronic equipment among the players in the chains. Article 33 of the law stipulates the obligation of manufacturers, distributors, traders, recyclers, and importers to structure their reverse logistics systems for the return of post-consumer WEEE [2]. Reverse logistics as a process aims to provide a destination for the return of the WEEE in the business cycle in an environmentally correct way, specifically one that is supported by legal terms.

It should be noted that, in October 2019, a sectoral agreement was also signed, which aimed at sharing the responsibilities for WEEE management between manufacturers, waste managers, and recyclers in Sao Paulo, Brazil. Oliveira Neto et al. [3] mention that electronics manufacturers are responsible for implementing post-consumer WEEE reverse logistics, whereby WEEE circularity through remanufacturing, repair, reuse, recycling, and/or sale to the secondary market is aimed. However, the manufacturers cannot simultaneously carry out production and remanufacturing. Thus, a common strategy has been to hire a WEEE manager who is responsible for allocating collection points and reverse logistics, as well as for receiving, disassembling, segregating, and disposing of appropriate recyclers and then reselling these on the secondary market.

Thus, it is considered relevant to incorporate the principles of the circular economy in the reverse logistics operations associated with WEEE [4], namely to consider the players in the reverse chains [5] and to establish the circularity of WEEE through remanufacturing, repair, reuse, recycling, and/or sale to the secondary market. This is to be performed [3] in compliance with the regulation of the local policies that are aimed at eliminating the disposal of WEEE in landfill, thus allowing for a reduction in CO_2_ in addition to improving the recovery rate, as well as the capacity of the facilities in relation to the total expected profit [6]. By implementing cyclic material flows, it is possible to limit the production flow to levels that nature tolerates, thus respecting its natural reproduction rates [7] and generating system sustainability [8].

To promote the implementation of the principles of the circular economy in the reverse WEEE chains—one that is stimulated by the mandatory adoption of reverse logistics in Sao Paulo by the electronics sector—it is timely and relevant to conduct studies that can augment the knowledge about how to improve such systems [3]. In this context, it is important to simulate the economic and environmental optimization of the WEEE reverse network (manufacturers, waste managers, collection points, and recyclers in Sao Paulo). This can be obtained from the application of simulation and artificial intelligence technologies in optimizing the routing of vehicles that are used in waste collection. Simulation can be considered an Industry 4.0 technology, and it is used for real-time data analysis, for offering opportunities for adjustments in complex systems through knowledge, and for information and accurate estimates about the system [9]. It is important to adopt information technology infrastructure [10]. The use of simulation and artificial intelligence approaches with a focus on the circular economy for the economic and environmental optimization of reverse chains of WEEE can generate economic and environmental benefits in operations [11].

A systematic literature review was carried out addressing 23 scientific papers, which are presented in Table 1. These were selected to investigate the use of simulation and artificial intelligence approaches as ways through which to optimize the WEEE reverse logistics network, both from economic and environmental points of view. The first step of the analysis involved the characterization of the computational techniques that were applied in each selected work. Among these, nine studies used mixed-integer linear programming for the simulation task—as shown in Achillas et al. [12]; Qiang; Zhou [13]; Kilic et al. [14]; Bal; Satoglu [15]; Elia et al. [16]; Mar-Ortiz et al. [17]; Gomes et al. [18]; Alumur et al. [19]; and Assavapokee and Wongthatsanekorn [20]. Three other studies used multicriteria objective linear programming—as seen in Achilles et al. [21]; Achillas et al. [22]; and Yu and Solvang [23]. Also, two studies dealt with discrete event simulation—Gamberini et al. [24] and Shokohyar and Mansour [25]. One piece of research adopted linear and nonlinear optimization methods in discrete or continuous variables, i.e., Dat et al. 2012 [26], and another study used stochastic programming, i.e., Ayvaz et al. [27]. Some studies combined simulation and artificial intelligence techniques, as was found in the nonlinear gray Bernoulli model with the convolution integral NBGMC that was improved by Particle Swarm Optimization in Duman et al. [28], as well as in the multi-objective models that were computed using the two-phase fuzzy compromise approach developed by Tosarkani et al. [6]. In addition, Lv and Du used the Kriging method [29], whereas Moslehi et al. [30] used a multi-objective stochastic model and a bi-objective mixed-integer programming model under certain uncertainties. An approach using system dynamics and a mixed-integer nonlinear programming model was utilized by Llerena-Riascos et al. [4], and a convolutional neural network-based quality prediction and closed-loop control method was used by Zhang et al. [31]. Lastly, agent-based modeling, system dynamics, and discrete event simulation were implemented by Guo and Zhong [5]. 

After analyzing the published works, the first gap identified was that no research applied artificial intelligence (genetic algorithms combined with Clarke & Wright heuristic) for the economic and environmental optimization of the reverse WEEE network, considering manufacturers, waste managers, and recyclers. It should be noted that the choice of techniques to compose the proposed simulation model to optimize routes was because both have shown good results (optimal or suboptimal solutions) in routing and other related problems of combinatorial optimization. In addition, they also allow handling with many constraints, including time windows and capabilities present in the addressed problem, as shown in Koç et al. [32]. Nevertheless, the use of heuristic algorithms, such as Clarke & Wright [33], to generate the initial population of the genetic algorithms with feasible solutions has proven to be a good alternative for solving routing problems, as demonstrated in the works of Lima et al. [34] and Lima & Araujo [35]. 

A second step in the research analysis was to understand the procedure used to present the environmental gains in the selected studies. In this regard, the studies that presented an environmental assessment—listed in Table 1—only offered a quantification in percentage using data extracted from the computer simulation. These studies generally emphasized the reduction of WEEE disposal in landfills and the reduction of CO_2_, such as in Gamberini et al. [24]; Achilles et al. [21]; Achilles et al. [22]; Assavapokee and Wongthatsanekorn [20]; Shokohyar and Mansour [25]; Yu and Solvang [23]; Bal and Satoglu [15]; Elia et al. [16]; Duman et al. [28]; Tosarkani et al. 2020 [6]; Llerena-Riascos et al. 2021 [4]; Lv and Du [29]; Moslehi et al. [30]; and Guo and Zhong [5]. From this, the second gap identified was that no research evaluated the reduction of environmental impacts in the abiotic, biotic, and water dimensions using the Material Intensity Factor (MIF), which is a relevant tool for global assessment of the minimization of environmental impacts, not just using percentage data.

Third, the procedure used by the surveys to develop the economic evaluation was analyzed. They use only percentage data to measure transportation and storage cost savings and profitability without a detailed explanation of the data. Thus, the third research gap consists of the fact that no research was identified presenting the calculation of cost and time reduction in detail, in addition to measuring the improvement in the volume of vehicles, being a primordial aspect in the optimization and orientation of transport managers. It should be noted that complex optimization scenarios without detailing make it impossible for managers to apply them in practice.

The fourth aspect analyzed in the research was the country of application of computer techniques for optimization of the WEEE reverse chains. Five surveys were carried out in China by Dat et al. [26], Qiang and Zhou [13], Yu and Solvang [23], Lv and Du [29], and Guo and Zhong [5]; three surveys were conducted in Greece by Achillas et al. [12], Achillas et al. [21], and Achillas et al. [22]; three in Türkiye by Ayvaz et al. [27], Kilic et al. [14], and Bal and Satoglu [15]; two in Italy by Gamberini et al. [24] and Elia et al. [16]; two the USA by Assavapokee and Wongthatsanekorn [20], and Duman et al. [28]; two in Iran by Shokohyar and Mansour [25], and Moslehi et al. [30]; and one study in each of the following countries: Spain, by Mar-Ortiz et al. [17]; Portugal, by Gomes et al. [18]; Germany, by Alumur et al. [19]; Canada, by Tosarkani et al. [6]; Colombia, by Llerena-Riascos et al. [4]; and Belgium, by Zhang et al. [31]. Thus, the fourth research gap was the lack of studies carried out in Brazil, mainly in Sao Paulo, on the use of simulation and artificial intelligence for economic and environmental optimization of the reverse WEEE network, considering manufacturers, waste managers, collection points, and recyclers.

The fifth point analyzed concerned the use of the circular economy approach in research. Tosarkani et al. [6] mentioned the circular economy only in terms of the passage of Bill 151 and the development of circular economy strategies in Ontario, Canada with greater attention directed towards recycling electronics (OES annual report, 2017). Llerena-Riascos et al. [4] incorporated the principles of the circular economy to maximize economic benefits concerning environmental ones, and Guo and Zhong [5] mentioned the circular economy superficially, in addition to not performing optimization in the WEEE reverse chains. Therefore, no research on the circular economy was identified that evaluated the environmental impact and economic gain with details of the adoption of computer simulation and artificial intelligence for optimization of the WEEE reverse chains, denoting the fifth gap explored in this study. Hidalgo et al. [11] concluded that the use of simulation for the management of WEEE in the reverse network does not guarantee eco-efficiency because it is important to use the circular economy approach to obtain economic and environmental benefits in operations since it is about promoting the circularity of WEEE.

Therefore, the application of simulation and artificial intelligence for economic and environmental optimization of the reverse WEEE network, considering manufacturers, waste managers, and recyclers to promote circular economy, was not addressed in the scientific literature. The exploratory analysis of the literature allowed the identification of five important research gaps, which are critical for both theory and managerial practice, and from which the following research question was formulated: How can simulation and artificial intelligence techniques be applied for economic and environmental optimization of the reverse WEEE network (manufacturers, waste managers and recyclers of Sao Paulo), to promote circular economy?

This study contributes to filling the identified research gaps, suggesting that an optimized configuration of the reverse WEEE network should promote economic gains and the reduction of environmental impacts for the actors in the network (manufacturers, waste managers, and recyclers in Sao Paulo). The economic gains are evaluated considering the opportunity to reduce transport costs. The reduction of environmental impacts is measured by evaluating the intensity of the material in the abiotic, biotic, water, land, and air compartments in terms of the minimization of fuel consumption and CO_2_ emission. Therefore, this study is justified by its contribution to theory, organizational practice, and society.

## 2. Systematic Literature Review of the Simulation Approach for Optimizing the WEEE Reverse Logistics Network

A preliminary overview of the extant literature is organized in Table 1, displaying the research work that used computer simulation to identify the optimization of the WEEE reverse logistics network in economic and environmental terms. Three surveys were carried out in Greece, three in China, three in Turkey, two in Italy, and one each in Spain, Portugal, Germany, the USA, and Iran. All surveys aim to reduce transport costs, and half aim to reduce CO_2_ emissions. Likewise, in China, an additional five studies were identified. Dat et al. [26] developed a model to minimize the total cost of the WEEE recycling network in China, which is the sum of the transport cost, operation cost, fixed cost, and disposal cost minus the revenue generated from the sale of recyclable materials. and renewables and components. Also, based on the proposed model, the ideal locations of the facilities and the material flows in the reverse logistics network were determined. Qiang and Zhou [13] developed a robust mixed-integer linear programming simulation model for the WEEE reverse logistics network to optimize the handling process, which was affected by recovery uncertainty based on risk preference coefficient and risk coefficient penalty diverted from restrictions that could allow decision makers to fine-tune operating system robustness and risk preferences. The result showed an opportunity to reduce transport costs.

Yu and Solvang [23] developed a stochastic mixed-integer programming model to design and plan a multi-source, multi-echelon, capable, and sustainable reverse logistics network for managing WEEE under uncertainty. The model considers economic efficiency and environmental impacts in decision-making, and environmental impacts are evaluated in terms of carbon emissions. Lv and Du [29] developed a simulation based on the Kriging method to predict the amount of WEEE returned in reverse logistics in China. The proposed model can accurately predict the amounts of WEEE returned from unknown locations, as well as those from the entire area, through data from the known location, which is important for compliance with environmental legislation. Guo and Zhong [5] applied agent-based modeling, system dynamics, and discrete event simulation constructed in China for the simulation of a closed-loop supply chain based on the Internet of Things, allowing the generation of more profit and reducing more greenhouse gas emissions and contamination by heavy metals, in addition to offering protection of people from diseases caused by heavy metals present in WEEE. It should be noted that this study mentions adequate management of WEEE using IoT, installation of sensors in a truck, barcodes on products, trash cans on the sidewalk, and a pre-selection center, in addition to the technology adopted by the manufacturer. However, this study does not perform optimal optimization and mentions circular economy only superficially.

Three works in Greece were identified. Achillas et al. [12] used the mixed-integer linear programming model to adapt the model to optimize and minimize total costs of transporting WEEE between collection points and recycling units, optimizing the use of containers and storage containers of WEEE, and cost reductions on WEEE storage deposits in Macedonia and Greece. This suggests that some network nodes (i.e., collection and recycling points) can be strategically modified in such a way as to promote considerable cost reductions in the WEEE reverse logistics network. Achilles et al. [21] adopted multicriteria objective linear programming to identify the optimal location for the installation of waste recycling plants for two cities in Greece: Messologhi and Kavala. The study aimed to address legal standards and goals for the collection of WEEE. Specifically, the goal was to minimize the environmental impact by reducing the possibility of WEEE being dumped in landfills, in addition to reducing pollutants from fossil fuels (CO_2_) in the atmosphere. Moreover, it documented the generation of an economic advantage of EUR 235,000 due to the recycling and reuse of WEEE and the minimization of fuel consumption. Achillas et al. [22] used multicriteria objective linear programming for the weighted optimization of WEEE collection and recycling processes to minimize total logistic costs and reduce fuel consumption in the region of Central Macedonia, Greece. The results showed a 5% reduction in CO_2_ pollutants (from fossil fuels), in addition to an economic gain of EUR 545,000.

Two surveys were carried out in Italy. Gamberini et al. [24] developed discrete event simulation and lifecycle analysis for the optimization of the WEEE transport network in northern Italy. The authors used vehicle routing methods and heuristic procedures for creating different scenarios for the system, simulation modeling to obtain solutions that satisfy technical performance measures, life cycle analysis to assess the environmental impact of such solutions, and multicriteria decision methods to select the best choice under the joint technical and environmental perspective. With this, opportunities to reduce transport costs were identified, in addition to minimizing CO_2_ emissions in the environment. Elia et al. [16] do not mention optimization considering the reverse chains in terms of recycling and reuse. They only mention collection and direct options or the determination of another path in Italy. With this, the simulation was adopted to compare different alternatives for a WEEE collection service. A dynamic collection scheme (i.e., with varying collection frequencies based on the actual level of waste stream) is simulated in two different logistical configurations, i.e., one based on direct connection and the other based on a network. The impact of the adoption of electric vehicles is also evaluated. Alternatives are compared using key economic and environmental performance indicators to assess the level of sustainability. The simulation was adopted to compare alternatives for a WEEE collection service. A dynamic collection scheme (i.e., with varying collection frequencies based on the actual level of waste stream) is simulated in two different logistical configurations, i.e., one based on direct connection and the other based on a network. The impact of the adoption of electric vehicles is also evaluated. Alternatives are compared using key economic and environmental performance indicators to assess the level of sustainability. A simulation was employed to compare different alternatives for a WEEE collection service. A dynamic collection scheme (i.e., with varying collection frequencies based on the actual level of waste stream) is simulated in two different logistical configurations, i.e., one based on direct connection and the other based on a network. The impact of the adoption of electric vehicles is also evaluated. Alternatives are compared using key economic and environmental performance indicators to assess the level of sustainability—one based on direct connection and one based on a network.

Assavapokee and Wongthasanekorn [20] used the mathematical model of mixed-integer linear programming for process optimization in terms of an adequate choice for the implantation of recycling units in Texas, USA through discrete variables, representing decisions such as locations and capacity allocation. Overall, they also addressed decisions about the material flows of the reverse logistics network. The model considered the obsolescence estimates for the products (e.g., computers, monitors, televisions) and the sales volume of the products, and analyzed the logistics transport costs. This reduced the logistics costs of transporting waste in terms of fuel and reduced storage costs for waste deposits, in addition to the percentage indication of the reduction in the amount of CO_2_. Other authors [28] proposed a nonlinear gray Bernoulli model with convolution integral NBGMC (1, n) improved by particle swarm optimization (PSO) with the aim of presenting a new prediction technique for e-waste for the US with multiple inputs in the presence of limited historical data. It was concluded that it is possible to improve decision-making in reverse logistics planning, allowing the proper collection, recycling, and disposal of electronic waste, generating elimination of WEEE disposal in landfills and CO_2_ reduction.

Shokohyar and Mansour [25] designed a WEEE recovery network to determine the best locations for collection centers and recycling plants for total WEEE management in Iran so that the government can simultaneously trade between environmental issues and economic and social impacts. Moslehi et al. [30] applied the multi-objective stochastic model and bi-objective mixed-integer programming model under uncertainties with the aim of modeling the reverse logistics process of electrical and electronic equipment (EEE) in Iran. A case study of an electronic equipment manufacturer in Esfahan, Iran, was included, making it possible to minimize the disposal of WEEE in landfill and reduce CO_2_, in addition to reducing transportation costs.

Mar-Ortiz et al. [17] developed a survey in Spain with the aim of optimizing the design of the WEEE logistics network. Thus, first, an installation location problem was formulated and solved using mixed-integer linear programming; in the second phase, a new integer programming formulation for the corresponding heterogeneous fleet vehicle routing problem is presented, and an economics-based heuristic algorithm is developed to efficiently solve the related collection routing problems; in the third phase, a simulation study of the collection routes is carried out to evaluate the overall performance of the recovery system. The results show a good performance of the proposed procedure and an improved configuration of the recovery network in relation to the one currently in use.

Gomes et al. [18] developed a generic mixed-integer linear programming model that was proposed to represent this network, which is applied to its design and planning in Portugal, where the best locations for the collection and sorting centers are chosen simultaneously with the definition of network tactical planning. Some analyses were carried out to provide more information on the selection of these alternative sites. The results support the strategic expansion plans of the companies for the opening of many centers and their location close to the main sources of WEEE, with a main focus on reducing operating costs (see Alumur et al. [19]).

Tosarkani et al. [6] applied efficient solutions of the multi-objective model using the two-phase fuzzy compromise approach, aiming to optimize and configure a Canadian WEEE reverse logistics network, considering the uncertainty associated with fixed and variable costs, the amount of demand and return and the quality of returned products. The study mentions that with the passage of Bill 151 and the development of circular economy strategies in Ontario, greater attention has been directed towards recycling electronics (OES annual report, 2017). With this, it is necessary to implement reverse chains for environmental compliance to reduce pollution and eliminate the disposal of WEEE in landfills, allowing the reduction of CO_2_, in addition to improving the recovery rate.

Llerena-Riascos et al. [4] applied simulation using system dynamics and mixed-integer nonlinear programming to design sustainable policies for WEEE management systems, incorporating circular economy principles to maximize economic benefits in relation to environmental ones. This led to a 33% increase in profit and a 65% increase in environmental benefits. It should be noted that despite mentioning the circular economy, it does not quantify the economic and environmental gains based on a real case, presenting percentage data.

Zhang et al. [31] applied computer simulation using convolutional neural network-based quality prediction and closed-loop control with the aim of presenting a closed-loop capture planning method for the random collection of WEEE products in Belgium, reducing costs in the collection and pre-processing process.

## 3. Methodology

The study builds a preliminary systematic literature review with the purpose of framing the investigated field and supporting the development of the conceptual model. After the literature review and analysis, interviews were conducted with managers of companies involved in the reverse chains of WEEE. After data collection, the proposed AI approach was developed for optimizing the routing of vehicles on the reverse WEEE network. Next, the environmental and economic assessments were carried out using the information derived from the routing. Based on the simulation with the proposed approach, a new interview process was carried out with the companies to investigate their opinions on the simulation and the optimization model developed.

### 3.1. Systematic Literature Review

A systematic literature review (SLR) was conducted to better understand the state of the art in terms of applying simulation and artificial intelligence techniques for optimizing routes aiming for economic and environmental gains in the reverse WEEE network. This step also guided the development of the items for the composition of the semi-structured interview questionnaire.

The keywords used for the development of this study and which were adopted in the systematic literature review are:(i)“simulation” AND “reverse logistics” OR “reverse chains” OR “closed-loop” AND “waste electrical” OR “weee” OR “electronic”.(ii)“modeling” AND “reverse logistics” OR “reverse chains” OR “closed-loop” AND “waste electrical” OR “weee” OR “electronic”.(iii)“genetic algorithms” AND “reverse logistics” OR “reverse chains” OR “closed-loop” AND “waste electrical” OR “weee” OR “electronic”.(iv)“artificial intelligence” AND “reverse logistics” OR “reverse chains” OR “closed-loop” AND “waste electrical” OR “weee” OR “electronic”.(v)“optimizing” AND “reverse logistics” OR “reverse chains” OR “closed-loop” AND “waste electrical” OR “weee” OR “electronic”.

As mentioned, these keywords were used to identify scientific research on the researched subject. A total of 31 articles were identified, out of which 16 were selected after the systematic review. The selection criterion adopted was that research should use simulation to optimize WEEE reverse logistics routes.

Based on these articles, the variables used for simulation and routing optimization were identified. 

The literature review also allowed for the development of a semi-structured questionnaire to support the conducting of interviews for data collection. The topic questions addressed in the semi-structured instruments were as follows:(a)General information about the companies;(b)Description of the reverse chain processes, which include the manufacturer, waste manager, and recyclers, in addition to their locations and exclusive specialties by type of electronic waste;(c)Identification of manufacturers, waste managers, collection points, and recyclers, as well as the amounts of electronic waste received per month/year, the number of materials and substances processed per month/year, and the total WEEE processing capacity per month/year.

### 3.2. Procedure for Data Collection—Expert Analysis via Semi-Structured Interview

In this research, semi-structured interviews were conducted with four manufacturers’ managers, a WEEE manager, and three recyclers located in Sao Paulo to survey the volume and types of WEEE, locations of recyclers, and collection points. Thus, the research method involved collecting information from specialists from companies belonging to the reverse WEEE network. Bogner et al. [36,37] mentioned that the analysis of specialists allows for verifying the researched subjects in the organizational practice, making it possible to gather important data to develop research with relevant practical contributions. However, for the effectiveness of the data-collection process with specialists, semi-structured interviews were carried out.

With this, it was possible to study and understand the reverse chain processes of WEEE, in addition to raising quantitative data on volumes and types of WEEE, and locations of companies in Sao Paulo, allowing the determination of the set of routes that optimize environmental and economic gains, promoting the circular economy.

### 3.3. Procedure for Data Analysis

The data collected about the recyclers and collection points were recorded in an Excel spreadsheet to facilitate its analysis using the AI approach. This, building on the optimization of the routes for the reverse WEEE chains, provides a comparative analysis of the economic/financial aspects and the environmental impact by evaluating the intensity of material in the compartments: abiotic, biotic, water, air, and land. With this, the optimal model of the WEEE reverse chains was determined in economic and environmental terms, which are important factors for business decision-making.

#### 3.3.1. Proposed AI-Based Approach for Economic and Environmental Assessment

The approach proposed in this study is focused on the optimization of routes in the reverse WEEE chains, which in turn directly impacts the reduction of economic and environmental costs. It is composed of a set of four computational pipelined procedures developed in the Python language, and its functioning is illustrated in Figure 1 and detailed below.

In the first step (geolocation procedure), the geopy (https://pypi.org/project/geopy/ accessed on 24 July 2023) library is employed to provide the geolocations of the recyclers (RECs) and the collection points (CPs) from their addresses stored in an Excel spreadsheet received from the Sao Paulo State Waste Management Company, which also includes other information such as capacities of processing and collecting of the RECs and CPs, respectively. This spreadsheet containing raw data is the main input of the AI approach.

It is important to highlight that the problem addressed in this study that involves three recyclers (RECs) for collecting waste from 554 collection points (CPs), from the combinatorial optimization point of view, is classified as the multi-depot vehicle routing problem (MDVRP) which is much more complex than a simple VRP or CVRP (capacitated vehicle routing problem) that considers only one depot. However, this problem can be solved in two phases (clustering and routing), greatly reducing its complexity since it is broken down into simple VRPs or CVRPs. As this strategy was adopted in this study, the data analysis and representation procedure that comprises the second step first groups the CPs according to their proximity to the RECs and then uses the geocoordinates of RECs and CPs to compute the distance and time matrices (one for each REC) using the open-source routing machine—OSRM (https://project-osrm.org/ accessed on 24 July 2023) library. From these matrices, the distances and times spent on routes taken by vehicles in waste collection are computed. The procedure of the second step also creates the representation of a CVRP scenario for each REC using the computed matrices (DM and TM) and other information from the georeferenced data spreadsheet. In the literature, a scenario of CVRP is known as an “instance” of CVRP, which can be mathematically represented by Equations (1)–(8). In short, solving CVRP means finding a set of routes, where each route is traveled by a vehicle, with the objective of minimizing the total cost of routes (*tc*), respecting the following restrictions: (i) each route must start and finish at the depot (or distribution center); (ii) each customer must be visited only once; (iii) the sum of demands of customers belonging to a route cannot exceed the capacity of the vehicle; and (iv) all vehicles of the fleet must have equal capacities.
(1)$$minimize\;tc=\sum_{i=0}^{nc}\sum_{\begin{array}{c} j=0\\ j\neq i \end{array}}^{nc}\sum_{k=1}^{K}c_{ij}x_{ijk}$$
(2)$$Subject\;to:\;\sum_{k=1}^{K}\sum_{\begin{array}{c} j=1 \end{array}}^{nc}x_{0jk}\leq K$$
(3)$$\sum_{j=1}^{nc}x_{0jk}=\sum_{\begin{array}{c} j=1 \end{array}}^{nc}x_{j0k}=1;\;\;k=1,\ldots ,K$$
(4)$$\sum_{k=1}^{K}\sum_{j=0}^{nc}x_{ijk}=1;\;\;i=1,\ldots ,nc$$
(5)$$\sum_{j=0}^{nc}x_{ijk}-\sum_{j=0}^{nc}x_{ijk}=0;\;\;k=1,\ldots ,K\;;\;\;\;i=1,\ldots ,nc$$
(6)$$\sum_{k=1}^{K}\sum_{i\in S}\sum_{j\in S}x_{ijk}\leq \left|S\right|-v(S),\begin{array}{ccc} \forall S & \subseteq & V\backslash \end{array}\{0\},\left|S\right|\geq 2$$
(7)$$\sum_{i=1}^{nc}d_{i}\sum_{\begin{array}{c} j=0\\ j\neq i \end{array}}^{nc}x_{ijk}\leq cv;\;\;\;k=1,\ldots ,K$$
(8)$$x_{ijk}\in \left\{0,1\right\};\;\;i=1,\ldots ,nc;\;\;j=1,\ldots ,nc;\;\;k=1,\ldots ,K$$
where $d_{i}$: demand of customer *i*; *k*: vehicle; *K*: set of vehicles; *S*: Set of customers; *nc*: number of customers; *v*(*S*): minimum number of vehicles to service *S*; *cv*: capacity of vehicles; $c_{ij}$: cost of the path from the customer *i* to customer *j*; *tc*: total cost of all routes; $x_{ijk}$: path from customer *i* to customer *j* with vehicle *k*. In our case, customers are the CPs, and their demands are represented by their capabilities to collect waste; each REC acts as a depot (or distribution center); the service refers to the collection of waste; *v*(*S*) is obtained by dividing the sum of the capabilities of all CPs associated with a REC by the capacity of the vehicle used by the REC for collection of waste; and *tc* is computed taking into account distance (in km) and travel time stored in DM and TM matrices.

Equation (2) ensures that *K* vehicles will be used starting from the distribution center, while Equation (3) guarantees that each route has its beginning and ending at the distribution center. Equation (4) defines that customers must be attended to exactly one time, and Equation (5) keeps the flow, ensuring that the vehicle arrives at a customer and out of it, preventing the route from ending prematurely. Equation (6) prevents the formulation of routes that do not include the depot. Restriction 6 ensures that the number of vehicles used to service the customers of set *S* is not less than *v*(*S*), and, finally, Equation (7) is used to formulate the vehicle capacity restriction. Equation (8) explains that the solution to the problem is a binary matrix.

In the third step (vehicle routing procedure), the routing for the three RECs from the scenarios (instances) provided in the second step is conducted, and a report summarizing the routing information is generated (a fragment is depicted in Figure 2).

For the routing task, we employed the genetic algorithms—GA ([38,39])—combined with the Clarke & Wright—CW—algorithms [33]. GA is a meta-heuristic technique based on the theory of species evolution, according to which individuals in a population that best adapt to the environment in which they live are more likely to survive and reproduce. The CW, one of the most known heuristics for VRP, consists of a saving algorithm that seeks to replace the most expensive paths within the routes with paths that represent lower costs. When two routes (0, …, *i*, 0) and (0, *j*, …, 0) can feasibly be merged into a unique route (0, …, *i*, *j*, …, 0), then a distance saving *s*(*i*, *j*) = *c*(*i*, *D*) + *c*(*D*, *j*) − *c*(*i*, *j*) is generated. The GA and CW algorithms (adapted from [40]) are presented in Figure 3.

In our implementation using OR-TOOLS (https://developers.google.com/optimization/introduction/python?hl=pt-br, accessed on 24 July 2023) and HYGESE (https://pypi.org/project/hygese/ accessed on 24 July 2023) libraries, the CW is applied to generate feasible solutions for composing the initial population of the GA, which represents possible solutions to the problem. It should be noted that the choice of using GA and CW to compose the optimization model to obtain better routes was because both algorithms have shown good results for routing problems and other related combinatorial optimization problems. Nevertheless, the use of heuristics to populate the initial GA population with feasible solutions has proven to be a good alternative for solving routing problems [34,41,42,43]. In addition, we tested other metaheuristics such as Simulated Annealing—SA [44,45,46] and Tabu Search—TS [45,46], but the results obtained in preliminary experiments do not indicate improvements in using them.

Finally, in the fourth step, called the data visualization procedure, all data produced by the routing task is spatialized in a map that can be visualized as a web page in any browser, as shown in Figure 4. In our implementations, we employed the Open Street Map—OSM (https://www.openstreetmap.org/ accessed on 24 July 2023)—as well as the OSRM and folium (https://python-visualization.github.io/folium/ accessed on 24 July 2023) libraries. The Google Maps platform is more present in the daily lives of the people and is one of the most used options; however, its use is conditioned to the contracting of its services through its API or restrictions on its use, limiting the available public service. Thus, the combination of OSRM with folium and OSM represents a free and open alternative source to Google Maps for distance calculation and spatial data visualization.

Finally, the algorithms were implemented and executed on the Google Collaboratory platform (https://colab.google/ accessed on 24 July 2023), using the resources provided by the free version.

#### 3.3.2. Procedure for Economic Evaluation

The economic assessment was based on the cost of transporting the set of WEEE reverse chain routes, as well as the economic gains from recycling and reusing WEEE.

Table 2 presents six categories of transport costs that arose in the data-collection process with the WEEE reverse chain specialists. The sum of these costs represents the transportation cost, according to Equation (9).
(9)CT=CC+CMO+CS+CD+CET+CM

#### 3.3.3. Procedure for Environmental Evaluation

For environmental assessment, the Mass Intensity Factors (*MIF*) tool will be used, which allows the calculation of the environmental impact [48,49,50]. The data will be evaluated based on the volume of reused/recycled WEEE and the fuel spent on transport for the set of WEEE reverse chain routes studied. The MIF considers the mass (*M*) of the residue multiplied by the Intensity Factor (*IF*), according to Equation (10):(10)MIF=(M×IF)

The MIF tool allows the measurement of the environmental impact regarding the consumption of abiotic, biotic, water, and air materials so that each of them is grouped in the form of a compartment [51,52,53]. The biotic compartment is related to the set of all living organisms derived from plants and decomposers, while the abiotic compartment is related to a set of non-living factors of an ecosystem active in the biotic environment, such as pressure, temperature, rainfall relief, among others [54].

In this context, it should be noted that the calculation of the reduction of the environmental impact is obtained by multiplying the factor of each abiotic (*w*), biotic (*x*), water (*y*), and air (*z*) compartment by the reused/recycled and minimized mass. Thus, it is possible to calculate the sum of each compartment, considering the Material Intensity per Compartment (*MIC*), according to Equation (11):(11)$$MIC(w)={IF}_{A}\left(w\right)+{IF}_{B}\left(w\right)+{IF}_{C}\left(w\right)+{\ldots +IF}_{N}\left(w\right)$$
where: ${IF}_{A}\left(w\right)$, ${IF}_{B}\left(w\right),{\;IF}_{C}\left(w\right)$ and ${IF}_{N}\left(w\right)$ represent the intensity factors of residues *A*, *B*, *C,* and *N*, respectively, in the compartment *w*.

With this, the sum is calculated for each compartment (abiotic, biotic, water, air), which represents the reduction of environmental impacts per compartment. Then, it is necessary to calculate the total amount of minimization of environmental impacts by adding all the MICs, arriving at the MASS Intensity Total (MIT), according to Equation (12).
(12)$$MIT=MIC\left(w\right)+MIC\left(x\right)+MIC\left(y\right)+MIC\left(z\right)+\ldots +MIC(n)$$

The Material Intensity Factors used in this study are shown in Table 3.

To calculate the reduction in the emission of the main greenhouse gases and material particles generated in the transport operation, Equation (13) is used.
(13)Emissions=kwh of motor×worked hours×gas indexes 

## 4. Results and Discussion

### 4.1. The Reverse Chains of WEEE in Brazil

After the interviews with four electronics manufacturers located in Brazil, it was found that they did not have information on the players in the reverse WEEE chains, which includes a survey of the volume collected, locations of recyclers, and collection points in the region of Sao Paulo. The manufacturers mentioned that after the approval of the sectoral agreement in 2019, requiring the implementation of WEEE reverse logistics, a joint decision was taken between the manufacturers on hiring a WEEE manager, mentioning that she would have all the necessary information. The manager mentioned that she was unable to produce and handle WEEE simultaneously due to a lack of operational capacity. Thus, it was necessary to focus on manufacturing, its core competence.

In this interview, it was identified that the WEEE manager hired three recyclers for dismantling, recycling, and preparation for the sale of WEEE to the secondary market. The WEEE manager also referred us to a manager from one of the recyclers, emphasizing that he knew the process, including the estimated values needed to carry out the simulation. Thus, consequently, an interview was conducted in Sao José dos Campos (R1).

In this interview, the locations of recyclers and collection points were verified, as well as the volume of WEEE collected, as shown in Table 4. R1 has a planned capacity of 20,000 tons/year and a realized capacity of 6000 tons/year. Recycler 2 (R2) is in Sorocaba, with a planned capacity of 2500 tons/year and realized capacity of 700 tons/year, and Recycler 3 (R3) is installed in Nova Odessa (R3), with a planned capacity of 3200 ton/year and realized capacity of 1000 ton/year. The location of 554 collection points was also identified, as well as the volume of WEEE collected, ranging from 8 to 60 tons/year, which are distributed throughout Brazil. In this context, knowledge of the WEEE reverse chains used in Sao Paulo is new to the scientific literature. Also, the practical structuring of this WEEE reverse chain through simulation will contribute to the strategic actions of the circular economy, which aims to reduce, recycle, reuse, and recover as much WEEE as possible. However, it will not be possible to promote a closed-loop system, an aspect encouraged by circular economy actions. Instead, it will be fully optimized where possible, with the application of computational techniques over time, always aiming for the most efficient solution. This finding denotes a relevant theoretical and practical contribution because the research by Guo and Zhong [5] carried out in China, Tosarkani et al. [6] in Canada, and Llerena-Riascos et al. [4] in Colombia only used the term “circular economy”, but did not show the application with details showing the environmental and economic benefits, as this study elucidates.

Another aspect observed was that the WEEE manager does not have its own fleet of trucks for collection. She outsourced the transport to a carrier specializing in reverse logistics. The required vehicle was a VUC truck with a capacity of 3 tons because most collections are carried out in the metropolitan region and cities at collection points (stores, malls, parks, large and small supermarkets, etc.), not allowing the circulation of trucks with capacities greater than 3 tons. This finding shows that the manufacturers outsourced the reverse logistics of WEEE to a waste manager, which considers three recyclers and 554 collection points located in Sao Paulo. Thus, the manufacturers considered WEEE reverse logistics as a support activity and therefore outsourced it, innovating the state of the art.

However, even when interviewing the WEEE manager and recyclers, it was not possible to identify the current scenario with the details of the routes carried out by the VUCs, denoting the lack of global knowledge of the process, which could lead to future problems. In addition, the lack of knowledge of the environmental and economic gains of this action is related to the circular economy strategy.

In this context, the interviewee mentioned that the carrier performs 286 collections per month with 13 VUCs, considering 22 working days, including vehicle rotation, totaling 21,005 km driven. R1 performs 132 collections of 500 tons per month with 12,974 km. R2 performs 76 collections of 58 tons per month, driving 1930 km. R3 transports 83 tons through 53 collections over 6101 km. This information formed the current scenario, which considers the three recyclers and 554 collection points (Table 5). Thus, based on this information, a computational tool was applied to simulate the WEEE reverse chains, making it possible to present the optimization in economic and environmental terms.

### 4.2. WEEE Reverse Chains Simulation for Economic and Environmental Optimization

Initially, location information was entered for the purposes of plotting the map, and the association of each collection point to one of the three recyclers was assigned by considering the shortest distance between them, respecting the limit of the recyclability capacity of each recycler and the capacity of the vehicle for transport. Thus, if a collection point is closest to a recycler and it has reached maximum capacity, then that collection point is associated with the recycler with the second shortest distance.

The map of the State of Sao Paulo shows three recyclers in green, being related to R1 of Sao José dos Campos with 270 collection points (blue), R2 of Sorocaba with 164 collection points (black), and R3 of Nova Odessa with 120 points of collection (red), as shown in Figure 5. It should be noted that distance and real times computed using OSRM were considered. This finding is also innovative for research, similar to Achillas et al. [12], Achilles et al. [21], Achilles et al. [22], Ayvaz et al. [27], Kilic et al. [14], Duman et al. [28], Tosarkani et al. [6]. Research on the subject usually presents complex optimization scenarios directly using artificial intelligence, but it often lacks detailed explanations of knowledge construction. Furthermore, it typically fails to mention that the adoption of WEEE reverse logistics constitutes a strategic action at the level of the circular economy. Consequently, an operations manager could easily comprehend the optimization process, facilitating its replication in organizational practice and contributing to actions aimed at reducing, recycling, and reusing materials within the circular economy.

Table 5 displays the comparison between the current scenario and the optimized scenario. The current scenario was identified in the interview, as mentioned earlier, while the optimized scenario was extracted from the simulation using AI. It was found that it was possible to optimize the number of collections from 286 to 220, considering 642 tons per month. This reduced the distance covered by 26.87%, from 21,005 km to 15,360 km in the WEEE collection process at collection points for recyclers. It should be noted that the solution to the problem investigated in this study was through the meta-heuristic model, which considers the process of evolutionary computation of economies and minimization of environmental impacts. These savings represent how much distance or transportation costs can be relatively optimized and reduced, grouping themselves to the nodes of the networks and their respective destinations. The reduction of the environmental impact represents the route that consumes less fuel, as well as reuses more WEEE, reducing environmental impacts in the abiotic, biotic, water, and air compartments. An important finding was that the carrier fulfilled individual orders without a schedule for WEEE removal. Thus, the trucks returned most of the time with space to store more WEEE. In the optimization process, it considered the opportunity to develop the collection schedule, where, in each collection process, the VUC passes through several points until it loads the truck as much as possible. With this, this study presents the details of the researched scenario, contributing to the literature and organizational practice,

Thus, the reduction of 13 VUCs to 10 VUCs for the operation, considering that to collect 642 tons of WEEE per month in 22 working days, it is necessary to collect 29 tons per day in 10 VUCs of 3 tons each. As a result, it generated a reduction of 67,738 km driven per year, representing a gain of 26.87%, and a reduction in operating time from 315 h 5 m 56 s to 240 h 23 m 31 s, considering the operation of the WEEE reverse chains, generating savings of 74 h 42 m 2 s. It should be noted that the optimization sought to reach 220 h of operation to avoid overtime for drivers, but in the metropolitan region of Sao Paulo, this was not possible due to traffic. This result is important for the theory because in optimizing a realistic scenario in a large metropolis, such as Sao Paulo, it is not possible to optimize 100% of overtime in transport, i.e., it is not possible to program the route without traffic. It is noteworthy that this finding was not demonstrated in any study on the subject, denoting innovation in terms of managerial implications for the WEEE reverse logistics process.

Also, after optimizing the reverse WEEE chains in Brazil, the VUC occupancy rate improved from 74.82% to 97.21%, even using three fewer VUCs. This finding can be explained by considering that each collection process of each VUC passes through several collection points, making it possible to use the capacity of the vehicle better. Research on the subject indicates the percentage data of optimizations without detailing the reasons, e.g., detailing the reduction in the volume of VUCs. This aspect is strategic for the transport industry, being a relevant result for organizational practice, which also contributes to the circular economy. The reverse chain optimization reduced 3 VUCs, in addition to reducing the need to work a lot of overtime, due to better scheduling of collections at different points, simultaneously respecting the use of VUC cubage with better effectiveness. This study contributes to the theory because we are not aware of any research that applied simulation and artificial intelligence techniques for economic and environmental optimization of the reverse WEEE network in Brazil, specifically in Sao Paulo, considering the manufacturers, waste managers, and recyclers.

#### 4.2.1. Economic Gain with WEEE Reverse Chain Optimization

The WEEE manager is responsible for the operation of the reverse chains, which decided to outsource transport for the collection of WEEE at the 554 collection and unloading points at the three recyclers. An average value of USD 109.81 per freight was agreed with the carrier, and in the current scenario, 260 VUCs are leaving for monthly WEEE collection, totaling a monthly cost of USD 28,550.60 and annual cost of USD 342,607.20. It should be noted that a profit of approximately 25% was agreed with the carrier on operating costs, which in the current scenario is USD 274,126.80 and monthly is USD 22,843.90.

Table 6 shows the cost assessment of the current and optimized scenario. With the optimization of the WEEE reverse chain transport, the distance traveled per year was minimized from 252,058 to 184,320 km, saving USD 31,623.35 in fuel consumption.

It also reduced employee costs by USD 5661.82 due to the optimization of VUCs from 13 to 10 and the minimization of overtime by USD 3261.82 per year. This finding contributes to social gain because, due to better route planning, drivers spend less time in traffic or waiting to collect WEEE at different collection points. The social gain is related to a better quality of life for drivers, who will be able to work without overtime, making better use of time spent with their families. This result is a relevant aspect to promote the circular economy.

Also, with the optimization of the fleet, the following reductions per year were insurance costs of USD 7200.00, depreciation cost of USD 17,997.75, costs with charges and fees of USD 1560.00, and maintenance costs of USD 5999.25. This is the first study that presents the details of the cost evaluation between the current and the optimized scenario, making it possible to clearly present the cost reduction with the optimization of the WEEE reverse chains. Research on the subject mostly presents total cost reductions without much detail, as is the case of research by Achillas et al. [22] held in Greece with the economic gain of EUR 545,000 (Mar-Ortiz et al. [17]) with a 29.2% reduction in transport costs in Spain; and Llerena-Riascos et al. [4] generating a 33% increase in profit in Colombia.

Based on this assessment, it was possible to require the carrier to reduce the operation costs to USD 255,105.775 per year and monthly to USD 21,258.82. This result shows that it is important to provide contractual transparency in terms of operating costs between the contractor (WEEE manager) and the contractor (carrier). It also shows that it was important to add to the contract that the gain of the carrier would be 25% on operating costs.

#### 4.2.2. Environmental Gain with WEEE Reverse Chain Optimization

Table 7 shows the elements divided into fuel/lubricant and the main components of the tire, which were optimized in the WEEE reverse logistics operation. Thus, there was a reduction of environmental impacts in the compartments: (i) abiotic in 33,621.92 kg, which represents factors related to global warming, air quality, and minimization of pollution in fauna and flora. It should be noted that these factors directly affect the health of society, increasing the social cost; (ii) water in 295,268 kg, reducing pollution in the local water system; and (iii) reduction in air pollution by 74,194 kg. With that, it generated a global minimization of 403,083.68 kg. In this context, this is the first study that calculates the reduction of environmental impacts in abiotic compartments, water, and air due to the optimization of the WEEE reverse chain transport using artificial intelligence. For the environmental assessment, the Material Intensity Factor was used, which is a relevant tool for the global assessment of the minimization of environmental impacts, not using only percentage data. For example, Achillas et al. [22] mentioned a 5% reduction in CO_2_ pollutants (from fossil fuels), and Llerena-Riascos et al. [4] reported that it generated 65% in environmental benefits. Thus, the research does not present the reduction of impacts in the abiotic compartments, water, and air, which are subjects not explored in the scientific literature in the published simulation models. This result contributes to the adoption of a circular economy due to the reduction of environmental impacts on transport.

Another relevant finding is that the tire components (rubber, steel, sulfur, and polyester) are the most relevant in the optimization, minimizing 41,411 kg, followed by the reduction of environmental impact due to diesel optimization in 321,998.80 kg. It should be noted that carbon black was not considered in this calculation because tire wear generates carbon particles, which are more related to dust that affects breathing, being classified as emissions, which will be calculated later. Thus, by optimizing the use of VUCs, tires, and fuel, consumption was further minimized, an innovative aspect in research that adopted optimization with the use of artificial intelligence, which in most cases is concerned with the tool used and not with reducing the impact of realistic environmental issues that promote the circular economy.

There was also a reduction in emissions of 45,493.25 kg of the main polluting gases resulting from the transport operation in Sao Paulo, as shown in Table 8. Carbon dioxide is the main pollutant of Earth’s atmosphere, which affects the health of society; however, 45,367.81 kg were optimized with the reduction of collections from 286 to 220, denoting an important contribution despite not solving the problem in its entirety. This finding innovates the state of the art by measuring the reduction of emissions of the main gases responsible for the greenhouse effect due to the optimization of the transport of the reverse chains of WEEE to promote a circular economy, as well as indicating to managers an important way to reduce gas emissions, mainly CO_2_, to contribute to the 2030 agenda.

Another interesting result was the reduction of particulate matter from 5.32 to 3.93 in air. Pollution from particles generated from carbon black from worn-out tires and gas emissions drastically affects the lungs, causing serious illnesses. The reduction generated was 1.39 kg per year, showing that the adoption of artificial intelligence for route optimization is a promising tool for reducing emissions of greenhouse gases and particles, which contributes to the circular economy, denoting a relevant theoretical and practical contribution. Table 8 considers for the calculation: kwh—75.7565; hours worked in current scenario/year—3782; carbon black—121; hours worked in the current scenario/year—2884.

## 5. Conclusions

After conducting the study, it was found that the optimization of Waste Electrical and Electronic Equipment (WEEE) reverse chains in Sao Paulo using artificial intelligence led to both economic and environmental benefits, therefore promoting a circular economy. The optimized scenario effectively reduced the number of collections, considering the numerous collection points the VUC passes through, thus maximizing its capacity.

### 5.1. Theoretical Recommendations/Implications

This study contributes to the theory by presenting the current functioning of WEEE reverse chains in Sao Paulo, as well as presenting the optimized scenario with details after the adoption of the proposed artificial intelligence approach. In the current operation, manufacturers outsourced WEEE management and reverse logistics operations, denoting that WEEE management using reverse logistics is a support activity. Also, with the application of artificial intelligence, it was possible to present an optimized scenario that considers the programming for the removal of WEEE, where, in each collection process, the VUC passes through several points until it loads as close as possible to its maximum capacity. This optimized WEEE reverse logistics schedule reduced the number of collections—3 VUCs and overtime—improving the quality of life for drivers, generating economic and environmental benefits, and promoting the circular economy. It should be noted that this is the first study that applied artificial intelligence using genetic algorithms for economic and environmental optimization of the reverse WEEE network, considering manufacturers, waste managers, and recyclers.

Therefore, the cost reduction was concluded by comparing the current scenario and the optimized one, forcing the contractor to reduce the freight price. This study is the first to calculate the reduction of environmental impacts in abiotic compartments, water, and air on the subject, mainly evaluating the minimization of emissions of the main polluting gases generated by trucks, contributing to circular economy actions. Thus, this is the first study that presents the calculation of cost and time reduction in detail, in addition to measuring the improvement in the volume of vehicles, which is a primordial aspect in the optimization and guidance of operations managers.

### 5.2. Organizational Practice Recommendations/Implications

The study also contributes to organizational practice because it organized WEEE reverse chain operations, an aspect that it was not possible to identify in the current scenario with operational details. It should be noted that complex optimization scenarios without detail are impossible for managers to apply in practice. Thus, an operations manager could easily understand the process performed for optimization, enabling its replication in practice. The main problems found in practice that can be solved with this optimization are: (i) not answering requests for individual collections but carrying out programmed collections at several collection points, taking advantage of the volume of the vehicle as much as possible; (ii) it was not possible to optimize 100% of the extra hours of the transport operation due to traffic, but it was possible to optimize as much as possible; (iii) the lack of vision of the whole WEEE reverse chains made it difficult to study the reduction of 3 VUCs in operation, generating environmental and economic gains. Another important aspect was the development of the contract with transparency in terms of costs, considering that cost reduction would lead to a reduction in freight price and environmental impact and the minimization of greenhouse gas emissions in a realistic scenario.

Also, this topic is relevant and emerging in the business environment of the electronics sector because a sectoral agreement was signed in October 2019 based on law 12,305 enacted in 2010 on the mandatory management of WEEE through reverse logistics aimed at sharing responsibilities for WEEE management between manufacturers, waste managers, and recyclers in Sao Paulo, who are structuring the reverse chains.

### 5.3. Social Recommendations/Implications

The implementation of the reverse WEEE network in Sao Paulo also contributes to society because the environmental impact resulting from the inadequate disposal of WEEE in common sanitary landfills and the generation of greenhouse gas emissions is being minimized. Furthermore, several informal recyclers with the ability to be part of the reverse WEEE network will be formalized, generating employment, and informal collectors may have the opportunity for formalized employment.

The main limitation of this study was the realization of regional research in Sao Paulo, Brazil, justified because it is the first Brazilian state to implement WEEE reverse logistics. For future studies, this research should be carried out in other states and countries, with the aim of generating comparisons between them, making it possible to generate relevant results for theory, practice, and government actions. In addition, it is suggested that other AI techniques, such as the integrated multi-head dual sparse self-attention network proposed in [55], be used for solving learning problems related to the optimization of WEEE reverse chains.

## Figures and Tables

**Figure 1 sensors-23-09046-f001:**
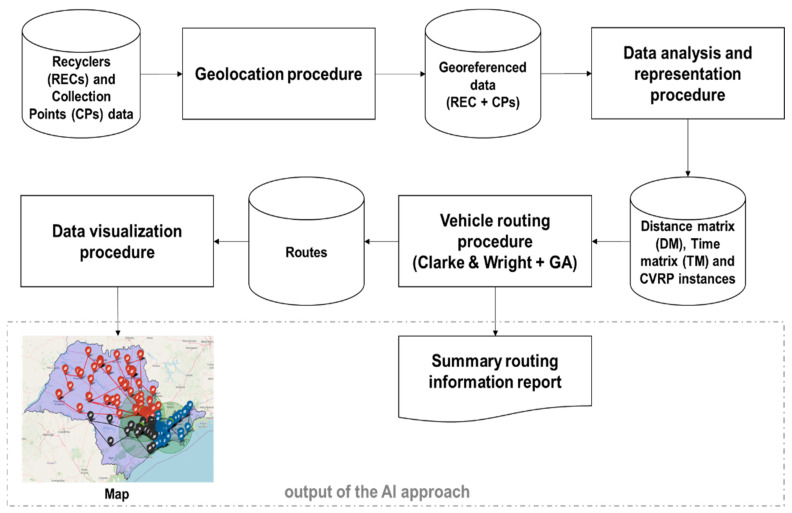
Schematic diagram of the working of the proposed AI approach.

**Figure 2 sensors-23-09046-f002:**
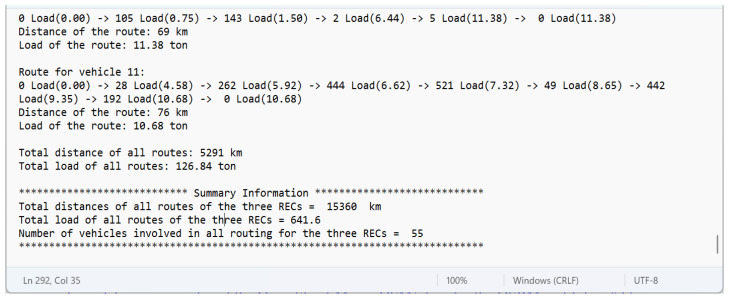
Fragment of summary routing report.

**Figure 3 sensors-23-09046-f003:**
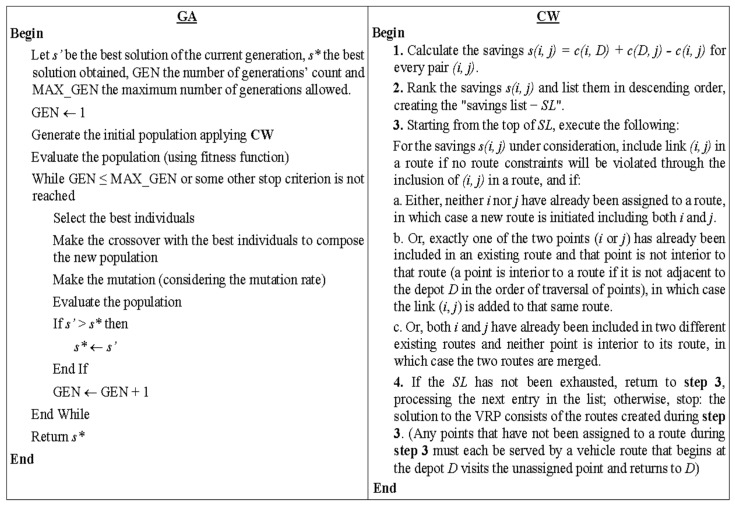
GA and CW algorithms.

**Figure 4 sensors-23-09046-f004:**
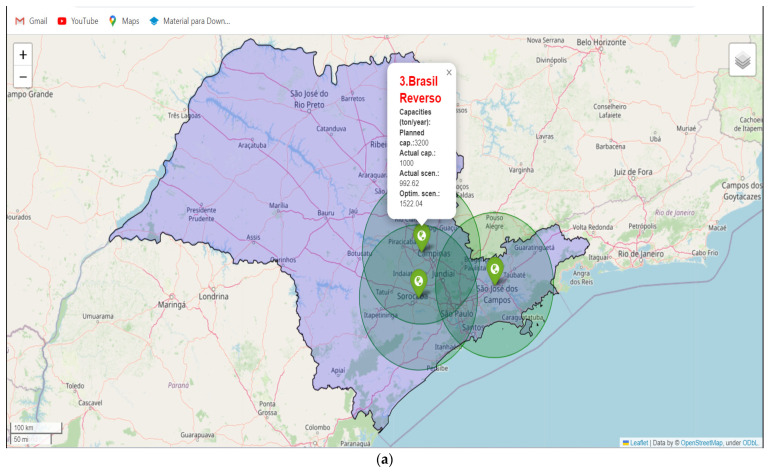
Data visualization maps. (**a**) only RECs, (**b**) RECs+CPs, and (**c**) RECs+CPs+routing.

**Figure 5 sensors-23-09046-f005:**
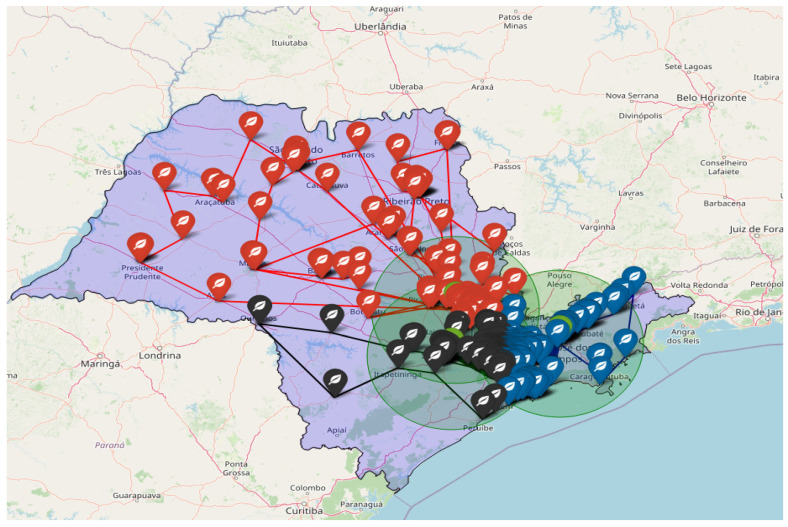
Map of Sao Paulo with the location and closest route between the three recyclers and 554 collection points.

**Table 1 sensors-23-09046-t001:** Research on simulation for optimization of the WEEE reverse logistics network.

Authors	Country	Computational Intelligence Used for Simulation	Mentions about EC	Aim of the Paper	Environmental Gain	Economic Gain	Social Gain
[26]	China	Linear and nonlinear optimization methods in discrete or continuous variables	null	Minimize the total cost of the WEEE recycling network		Reduction in the cost of transport and disposal, revenue generated from the sale of recyclable materials	
[13]	China	Mixed-integer linear programming	null	Optimize the WEEE reverse logistics network		Reduced transport costs	
[23]	China	Multicriteria for stochastic mixed-integer programming	null	Plan a reverse logistics network for managing WEEE under uncertainty	CO_2_ reduction	Reduced transport costs	
[29]	China	Model based on the Kriging method	null	Apply a spatial mathematical model based on the Kriging method to predict the amount of WEEE returns in reverse logistics	Environmental compliance		
[5]	China	Agent-based modeling, system dynamics, and discrete event simulation	yes	Establish a sustainable closed-loop supply chain system based on the Internet of Things, considering the economic, environmental, and social dimensions	CO_2_ reduction	Revenue generated from the sale of recyclable materials	Safety at work
[12]	Greece	Mixed-integer linear programming	null	Minimize total costs of transporting and storing WEEE between collection points and recycling units		Reduced transport costs	
[21]	Greece	Multicriteria objective linear programming	null	Identify the optimal location for installing waste recycling plants	Elimination of WEEE disposal in landfills and reduction of CO_2_	Recycling and reuse; reduction in fuel costs	
[22]	Greece	Multicriteria objective linear programming	null	Optimize WEEE collection and recycling processes to minimize total logistical costs and reduce fuel consumption	CO_2_ reduction	Reduced transport costs	
[27]	Türkiye	Stochastic programming	null	Minimize demand uncertainties for WEEE recycling by third-party recyclers to maximize profit		Reduced transport costs	
[14]	Türkiye	Mixed-integer linear programming	null	Design a WEEE reverse logistics system network structure		Reduced transport costs	
[15]	Türkiye	Mixed-integer linear programming and multi-facility, multi-product, and multi-period goal programming	null	WEEE collection process at service points, transport to recycling and waste recovery facilities	CO_2_ reduction	Reduced transport costs	job creation
[24]	Italy	Discrete event simulation and lifecycle analysis	null	Optimize the WEEE transport network	CO_2_ reduction	Reduced transport costs	
[16]	Italy	Mixed-integer linear programming	null	Compare different alternatives to a WEEE collection service	CO_2_ reduction	Reduced transport costs	
[20]	USA	Mixed-integer linear programming	null	Optimize processes in terms of the most appropriate choice for the implementation of recycling units in the network project	CO_2_ reduction	Reduced transport and storage costs	
[28]	USA	Proposed nonlinear gray model with convolution integral, improved by particle swarm optimization	null	To present a new prediction technique for multi-input junk e-mail predictions in the presence of limited historical data	Eliminate WEEE disposal in landfills and reduce CO_2_		
[25]	Iran	Discrete Event Simulation	null	Design a WEEE recovery network	CO_2_ reduction	Reduced transport costs	Employment generation, job security, local development
[30]	Iran	Multi-objective stochastic model. Bi-objective mixed-integer programming model	null	Model the electrical and electronic equipment (EEE) reverse logistics process as a bi-objective mixed-integer programming model under uncertainties	Eliminate WEEE disposal in landfills and reduce CO_2_	Reduced transport costs	
[17]	Spain	mixed-integer linear programming and heuristic algorithms	null	Optimize the design of the WEEE logistics network		Reduced transport costs	
[18]	Portugal	Mixed-integer linear programming	null	Optimize best locations for collection and sorting centers for reverse WEEE network planning		Reduced transport costs	
[19]	Germany	Mixed-integer linear programming	null	Maximize profit for WEEE reverse logistics network design problems		Reduced transport costs	
[6]	Canada	Multi-objective models are computed using the two-phase fuzzy compromise approach	yes	Optimize and configure an electronic reverse logistics network, considering the uncertainty associated with fixed and variable costs, the quantity of demand and returns, and the quality of returned products	Environmental compliance to reduce pollution. Eliminate WEEE disposal in landfills and reduce CO_2_	Revenue generated from the sale of recyclable materials	
[4]	Colombia	System dynamics and a mixed-integer nonlinear programming model	yes	To present an optimization-based simulation (OBS) approach that allows the design of sustainable policies for WEEE management systems	Environmental benefits	Revenue generated from the sale of recyclable materials	
[31]	Belgium	Convolutional neural network-based quality prediction and closed-loop control, named CNNB-CL	null	Closed-loop capture planning method is proposed for the random collection of WEEE products		Reduced transport costs	

**Table 2 sensors-23-09046-t002:** Transport costs.

Transport Costs (CT)	Concept
Fuel costs (CC)	The total amount spent on fuel and lubricants used to carry out the necessary displacement to meet the targets.
Labor costs (CMO)	The sum of the drivers’ gross wages, in addition to applicable surcharges, such as accommodation or bonuses.
Insurance costs (CS)	The total cost of insurance policies for both equipment and cargo, where applicable. In addition, premium payments and other charges should be included if necessary.
Depreciation costs (DC)	The total depreciation amount of equipment and accessories, including charges incurred to renew the vehicle.
Costs with charges and fees (CET)	The amount spent for the payment of taxes to enjoy the property—in this case, the equipment. However, costs with penalties and fines are also added to this item, in addition to toll costs.
Maintenance costs (CM)	The cost of carrying out preventive maintenance for equipment, such as revisions carried out by dealers or specialized technicians; however, the cost also includes unscheduled maintenance, for example, in the case of sudden breakdowns, as well as the cost of repairs due to small and medium accidents.

Source: Combes and Lafourcade [47].

**Table 3 sensors-23-09046-t003:** Material intensity factors.

Elements		Unid	Abiotic Material	Biotic Material	Water	Air
**Fuel and Lubrificant**	Diesel oil	Liters	1.36	0	9.7	3.2
	Engine oil	Liters	1.5	0	11.45	3.02
	Cooland fluid (Ethylene glycol)	Liters	2.9	0	133.46	2.29
	Oil for hydraulic system (Naphtha)	Liters	1.69	0	13.88	0.05
**Main tire components**	Polyisoprene Rubber	kg	5.7	0	146	1.65
	Steel	kg	9.32	0	81.86	0.77
	Sulfur	kg	0.25	0	4.1	0.7
	Polyester	kg	5.62	0	235.44	3.46

**Table 4 sensors-23-09046-t004:** Collection points and volume collected.

Types of Collection Points	Number of Collection Points	Total Collected Volume (tons/year)	Total Collected Volume (tons/month)
P—Batteries/pen drive/small computer eq	276	2484	207
PP—Batteries/Pen drive	91	764	64
G—Refrigerators/Stove/TVs/Washers/Air conditioning	30	1650	138
GG—Refrigerators/Stove/TVs/Washers/Air conditioning	13	770	64
M—Microwave/appliances	107	1712	143
Greenk—Computer EQ	15	135	11
Motostore—cell phone/computer eq	22	185	15
Total	554	7700	642

**Table 5 sensors-23-09046-t005:** Current and optimized scenarios.

Actual Scenary × Optimized Scenary													
						Actual Scenary (13 VUCs)				Optimized Scenary (10 VUCs)			
Recycler ID	Recycler	Planned Capacity (year)	Performed Capacity (year)	Planned Capacity (month)	Performed Capacity (month)	Total Distance Traveled (km)	Total Load (ton)	Collection Numbers	Time (s)	Total Distance Traveled (km)	Total Load (ton)	Collection Numbers	Time (s)
**GM&C**	**R1**	20,000	6000	1667	500	12,974	500	145	734,761	6339	326	112	381,099
**Sinctronics**	**R2**	2500	700	208	58	1930	58	82	115,536	3730	189	64	204,427
**Brasil Reverso**	**R3**	3200	1000	267	83	6101	83	59	284,059	5291	127	44	279,908
**Total per month**		**0**	**0**	**2142**	**642**	**21,005**	**642**	**286**	**1,134,356 s** **315 h:5 m:56 s**	**15,360**	**642**	**220**	**865,434 s** **240 h:23 m:31 s**
**Total per year**		**25,700**	**7700**	**0**	**0**	**252,058**	**0**	**3432**	**13,612,277 s** **3781 h:11 m:17 s**	**184,320**	**0**	**2640**	**10,385,208 s** **2884 h:36 m:48 s**
**Gain (Km) year**		**252,058 – 184,320 = 67,738**											
**Gain (Km %) year**		**184,320/252,058 = 26.87%**											
**Gain (h) month**		**1,134,356 s – 865,434 s = 268,922 s = 74 h:42 m:2 s**											
**Gain (h) year**		**13,612,277 s – 10,385,208 s = 3,227,069 s = 896 h:24 m:29 s**											
**VUC Capacity**		**3 ton per VUC**											
**Average loading percentage per month. The closer to 100% the use of each truck, the greater the economic and environmental gain.**		**Actual Scenary − 642 of total load ton/(3 ton of Vuc capacity × 286 of collection numbers) × 100 = 74.82%**											
		**Optimized Scenary − 642 of total load ton/(3 ton of Vuc capacity × 220 of collection numbers) × 100 = 98.56%**											

**Table 6 sensors-23-09046-t006:** Cost evaluation of the current and optimized scenario.

Actual Scenary (13 VUCs)					Optimized Scenary (10 VUCs)					Cost Reduction USD
**Fuel costs (FC)**	USD/year	Consumption	Price	Distance	**Fuel costs (FC)**	USD/year	Consumption	Price	Distance	**31,623.35**
	117,627.07	3 km/L	liter	km/year		86,003.71	3 km/L	liter	km/year	
		3	1.40	252,058			3	1.40	184,320	
**Labor costs (LC)**	USD/year	Salary	Extra Hour		**Labor costs (LC)**	USD/year	Salary	Extra Hour		**5661.82**
	14,552.73	800	1142h			8890.91	800	245 h		
			4152.73					890.91		
**Insurance costs (IC)**	USD/year	VUC			**Insurance costs (IC)**	USD/year	VUC	Carga		**7200.00**
	31,200.00	2400				24,000.00	2400	0		
**Depreciation costs (DC)**	USD/year	Price	Taxa		**Depreciation costs (DC)**	USD/year	Price	Tax		**17,997.75**
	77,990.25	39,995	15%			59,992.50	39,995	15%		
**Costs with charges and fees (CCF)**	USD/year	Price	VUC property tax	Others Tax	**Costs with charges and fees (CCF)**	USD/year	Price	VUC property tax	Others Tax	**1560.00**
	6760.00	39,995	320	200		5200.00	39,995	320	200	
**Maintenance costs (MC)**	USD/year	Price	Estimate		**Maintenance costs (MC)**	USD/year	Preço novo	Estimate		**5999.25**
	25,996.75	39,995	5%			19,997.50	39,995	5%		
**Annual Transport Cost (USD)**	**274,126.80**		Number of VUC	13	**Annual Transport Cost (USD)**	**204,084.62**		Number of VUC	10	**70,042.17**

**Table 7 sensors-23-09046-t007:** Material intensity assessment.

Elements		Unid (L, Kg)	Abiotic Material	Biotic Material	Water	Air	Material Intensity per Element
Fuel and Lubrificant	Diesel oil	22,580	30,708.8	0	219,026	72,256	321,990.8
	Engine oil	294	441	0	3366.3	887.88	4695.18
	Cooland fluid (Ethylene glycol)	252	730.8	0	33,631.92	577.08	34,939.8
	Oil for hydraulic system (Naphtha)	3	5.07	0	41.64	0.15	46.86
Main tire components	Polyisoprene Rubber	216	1231.2	0	31,536	356.4	33,123.6
	Steel	43	400.76	0	3519.98	33.11	3953.85
	Sulfur	35	8.75	0	143.5	24.5	1766.75
	Polyester	17	95.54	0	4002.48	58.82	4156.84
	* Carbon	121	null	null	null	null	0
Material Intensity per Compartiment (MIC)		null	33,621.92	0	295,267.82	74,193.94	null
**Material Intensity Total (MIT)**		**403,083.68**					
**Total Material Economized**		**23,561**					

* Carbon has no indicator, but it will be considered in the accounting calculation of gas reduction.

**Table 8 sensors-23-09046-t008:** Reduction in gas emissions.

Gases	Average Emission (g/kwh)	Actual Scenary (13 VUCs) Mass/Year (kg)	Otimized Scenary (10 VUCs) Mass/Year (kg)	Emission Reduction Mass/Year (kg)
CO	0.055	15.76	12.02	3.74
HC	0.023	6.59	5.03	1.56
NOx	1.746	500.25	381.47	118.78
CO_2_	666.886	191,070.23	145,702.42	45,367.81
Particule Material	0.018	5.32	3.93	1.39
**Emissions Total**		**191,598.15**	**146,104.86**	**45,493.29**

## Data Availability

Data are contained within the article.

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
