# Peer review of "Simulation of Electronic Waste Reverse Chains for the Sao Paulo Circular Economy: An Artificial Intelligence-Based Approach for Economic and Environmental Optimizations"

_sensors, 2023, doi:10.3390/s23229046_

Round 1

Reviewer 1 Report

The paper's composition is coherent; the structure is logical and meets the goal of the paper. The topic sounds very interesting. The title "Simulation of Electronic Waste Reverse Chain for São Paulo Circular Economy: an artificial intelligence-based approach for economic and environmental optimization" puts well the paper's objective; it is clear and expresses the issue being assessed very well. The abstract is formulated adequately along with the true picture of the paper. All the tools and methods the author uses are reasonable and well described and adequately fit the problem being assessed to give the reliable results. Conclusions are related to the results presented before reflecting the assessed issue at a professional level. All the tables are complete and understandable. Authors use enough tables, pictures and formulas featuring a great deal of data being processed hence adding a higher added value to the paper. However major revision would suffice to get the manuscript published in the journal. It is recommended that the authors make a relatively major revision, and the specific amendments to the text are as follows:

-         The goal explicitly stated within the Introduction clearly expressing the main problem and purpose of the paper and author's intention being assessed and discussed within the paper along with its clear and unambiguous formulation are required to be proposed.

-         Some rearrangement regarding the sections contents could be made. It is proposed to set aside the single units of the last chapter for Findings, Discussion and Conclusions.

-         The current Discussion is recommended to be moved to the Results and Findings section. Discussion section should be focused on some kind of polemic discourse comparing the research outcomes with the literature overview part that would be beneficial to be involved in Discussion.

-         Conclusion section is supposed to be separated from the last section; there could be provided some recommendations for practice based on the research outcomes as well as a final statement reflecting the assessed issue along with the way how the research results could be implemented in the practice bringing up any benefits and added value. The final research statements in Conclusion should be more supported by your evidence and arguments from your own research findings.

-         The English language style – spelling and grammar should be double-checked thoroughly.

Reviewer 2 Report

The following aspects should be addressed, before any further processing may take place.

1. The authors should compare, in a clearer manner, their contribution, to relevant existing approaches, and highlight their proposed approach merits and drawbacks. 

2. Thus, they should add a separate sub-section, in section 3 or in section 5, which should present the comparative analysis. The remarks should be justified with experimental data and/or meaningful conceptual remarks.

3. The MIF tool, and the experimental assessment, in general, should be clearly presented. The software and hardware specification should be described, the metrics should be clearly defined, and the numerical results should be clearly compared with the most relevant reference existing approaches.

4. Also, details concerning the software implementation of the proposed system should be provided (software libraries used, programming languages, etc.).

5. The English language should be improved through, at least, one round of proofreading, eventually using the help of a professional proofreader, or a native English speaker.

The English language should be improved through, at least, one round of proofreading, eventually using the help of a professional proofreader, or a native English speaker.

Reviewer 3 Report

This study contributes to the theory by presenting the current functioning of the WEEE reverse chain in São Paulo, as well as presenting the optimized scenario with details after the adoption of artificial intelligence. Overall, the paper is quite interesting. However, there are some problems in literature review, innovation, and comparison of results. Here are some specific suggestions:

1. It is recommended that the authors summarize the contribution points of the method proposed in this article in the introduction part of the article to highlight the research advantages of this article

2. Regarding the research overview of this article, some methods for AI analysis in this article are relatively old. It is recommended to analyze some recent work related to AI or DL, such as An integrated multi-head dual sparse self-attention network for remaining useful life prediction

3. On page 6 of the article, line 450, there may be problems with parameter interpretation

4. The theoretical basis of the article, such as formulas 1-8, etc., the methods described seem to be existing methods, so where is the unique innovation of this article?

5. Regarding the method analysis based on artificial intelligence, it is recommended to discuss some recently published learning methods at the method level, such as An integrated multitasking intelligent bearing fault diagnosis scheme based on representation learning under imbalanced sample condition

6. Some pictures in the article are not clear enough. It is recommended that the authors correct the picture format to increase the clarity of the picture

7. The description of the article is somewhat general, and the AI-based method used is somewhat generalized. What kind of AI method is used, what are the advantages, and where is the innovation reflected?

8. It is recommended to enrich the comparison of the results of the article and highlight the advantages of the research

Please see the comments to the authors.

Round 2

Reviewer 1 Report

The revised paper titled “Simulation of Electronic Waste Reverse Chain for São Paulo Circular Economy: an artificial intelligence-based approach for economic and environmental optimization“ intended to be published in Sensors (ISSN 1424-8220) Journal meets all the requirements for a professional scientific journal. All the significant comments, recommendations and remarks of reviewers have been incorporated into the manuscript in a proper way giving the paper higher added value and professional features.

Reviewer 2 Report

I appreciate that the changes made warrant further consideration of this paper.

English language is generally fine, I estimate that the English proofreading step would correct remaining issues.

Reviewer 3 Report

No further comments.

No further comments.